# Real-Life Performance of F-18-FDG PET/CT in Patients with Cervical Lymph Node Metastasis of Unknown Primary Tumor

**DOI:** 10.3390/biomedicines10092095

**Published:** 2022-08-27

**Authors:** Friederike Eilsberger, Friederike Elisabeth Noltenius, Damiano Librizzi, Joel Wessendorf, Markus Luster, Stephan Hoch, Andreas Pfestroff

**Affiliations:** 1Department of Nuclear Medicine, University Hospital Marburg, Philipps University Marburg, 35043 Marburg, Germany; 2Department of Otolaryngology, Head and Neck Surgery, University Hospital Marburg, Philipps University Marburg, 35043 Marburg, Germany

**Keywords:** FDG-PET/CT, unknown primary tumor, CUP, lymph node metastasis

## Abstract

Background: Neoplasms in the head and neck region possess higher glycolytic activity than normal tissue, showing increased glucose metabolism. F-18-Flourodeoxyglucose (FDG) positron emission tomography/computed tomography (PET/CT) can identify an unknown primary tumor (CUP). Aim: The aim of this study was to assess the real-life performance of F-18-FDG-PET/CT in detecting primary sites in patients with cervical lymph node metastasis of CUP. Methods: A retrospective data analysis of 31 patients who received FDG-PET/CT between June 2009 and March 2015 in a CUP context with histologically confirmed cervical lymph node metastasis was included. Results: In 48% of the patients (15/31), PET/CT showed suspicious tracer accumulation. In 52% of the patients (16/31), there was no suspicious radiotracer uptake, which was confirmed by the lack of identification of any primary tumor in 10 cases until the end of follow-up. FDG-PET/CT had a sensitivity of 67%, specificity of 91%, PPV of 92%, and NPV of 63% in detecting the primary tumor. Additionally, PET/CT showed suspicious tracer accumulation according to further metastasis in 32% of the patients (10/31). Conclusion: FDG-PET/CT imaging is a useful technique for primary tumor detection in patients in a cervical CUP context. Furthermore, it provides information on the ulterior metastasis of the disease.

## 1. Introduction

Most patients with cervical lymph node metastases suffer from a primary tumor in the region of the head and neck, although it is clinically non-detectable in some of these patients. These cases are referred to as carcinoma of unknown primary (CUP). The criteria for CUP syndrome are met when a histologically or cytologically confirmed malignancy is present without a demonstrable primary tumor after the completion of the primary diagnostic workup [1]. “Cervical CUP” is a relatively rare condition, representing between 2 and 9% of all cancers diagnosed in the head and neck, depending on the literature, and 70–90% of these patients are male [2]. The detection of the primary tumor in a CUP setting has extensive consequences for a patient’s therapy and prognosis [3]. In the diagnostic setting, conventional imaging consisting of magnetic resonance imaging (MRI) and computed tomography (CT) can be performed, as well as panendoscopy and functional imaging. The value of F-18-Flourdesoxyglucose (FDG) positron emission tomography/computed tomography (PET/CT) in the detection of the primary tumor in a cervical CUP setting is a subject of controversy in the literature. Some studies have demonstrated, in comparably sized patient samples, the performance of F-18-FDG-PET/CT for primary tumor detection in patients with CUP syndrome with cervical lymph node metastases. F-18-FDG-PET/CT can be used to identify an unknown primary tumor, since many neoplasms, including those in the head and neck region, possess high glycolytic activity and therefore demonstrate a high uptake of FDG. In view of the limited data available, the aim of our study is to demonstrate the value of F-18-FDG-PET/CT for primary tumor detection and the value of the additional information obtained in the real-life setting.

## 2. Materials and Methods

### 2.1. Patients

We retrospectively evaluated F-18-FDG-PET/CTs performed in the Department of Nuclear Medicine at University Hospital Marburg between June 2009 and March 2015 in a CUP context with histologically confirmed cervical lymph node metastasis (CLNM). Patients with lymphoma or previous carcinoma in the head and neck region that could be clearly related to cervical lymph node metastasis were excluded from this study. The study included 31 patients (Table 1).

In 25 patients (81%), the sampling date of the CLNM was known; in 16 patients (52%), the lymph node was removed by extirpation; a fine needle aspiration (FNP) was performed in six patients (19%); and a stamping biopsy was performed in three patients (10%). In six patients (19%), the sampling date could not be determined, and the examination was performed at the referring institution. At the point of time of PET/CT performance, the histopathological examination yielded, in 20 patients (65%), the unequivocal result of a metastasis of a squamous cell carcinoma (SCC); in eight patients (26%), it yielded a metastasis of an undifferentiated or not further characterized carcinoma. In one patient each, a metastasis of a malignant melanoma, a small-cell poorly differentiated carcinoma, and an adenocell carcinoma were diagnosed at this time.

All diagnostic procedures performed for the detection of the primary tumor or for the staging of the tumor disease between the diagnosis of CLNM and the F-18-FDG-PET/CT examination were recorded in this study. Nineteen of the thirty-one patients (61%) received a diagnostic evaluation to detect the primary tumor prior to PET/CT examination. In 48% of the patients (15/31), at least one “main diagnostic measure” (and possibly a minor diagnostic test) was performed for primary tumor detection or tumor staging prior to PET/CT examination. This included the following diagnostics: an MRI of the neck and a CT of the thorax and neck.

In 13% of the patients (4/31), only a “minor diagnostic test” had been performed prior to the PET/CT examination. These included the following examinations:

Panendoscopy, abdominal diagnostics (abdominal sonography, abdominal CT, diagnostic gastroscopy/colonoscopy), an X-ray of the thorax, MRI cranium, dermatological consultation, bronchoscopy, and CT cranium.

In 29% of the patients (9/31), the F-18-FDG-PET/CT examination was the first further diagnostic examination after a physical examination, ENT examination, and ultrasonography of the neck. In 10% of the patients (3/31), no statement could be obtained about which diagnostic procedure was performed because the diagnosis of CLNM was performed in a different hospital, and the patients were referred to the University Hospital Marburg for further treatment.

In 20 patients (65%), the PET/CT examination was performed within one month after the diagnosis of CLNM. Three patients (10%) received the PET/CT scan within one year of the diagnosis of CLNM. Two patients (6%) had longer intervals between the diagnosis of CLNM and PET/CT examination. In the remaining six patients (19%), precise information on the time interval could not be retrieved because the date of CLNM sample collection was unknown. The median time between diagnosis of CLNM and PET/CT examination was eight days (range: 1–3490 days).

Four of the patients (13%) received radiotherapy and chemotherapy before PET/CT examination. One patient (3%) received radiotherapy only. The remaining 26 patients (84%) did not receive any therapy before PET/CT examination.

Furthermore, 29/31 patients received panendoscopy: 1 ahead of F-18-FDG-PET/CT and 28 after PET/CT for diagnosis confirmation. We define panendoscopy according to the CUP scheme as microlaryngoscopy, esophagoscopy, tracheobronchoscopy, and epi- and oropharyngoscopy, with the collection of samples from the base of the tongue, epipharynx, and bilateral tonsillectomy. Panendoscopy according to the CUP scheme was performed on nine patients and revealed a tumor diagnosis in seven of these patients. In 16 patients, the collection of samples was not performed in concordance with the CUP scheme, but samples were also taken. In 11 of these patients, morphological abnormalities were found and biopsied, revealing the primary tumor in seven cases. In six patients, no samples were taken from the ENT area, although four of these patients received panendocsopy.

The patients were followed until June 2015, specifically to determine the courses of disease in patients without suspicious findings in the PET/CT. The average follow-up period (after PET/CT examination) of the patients was 1.9 years, and the median was 1.28 years (range: 0.04–5.39 years). Ten patients with a true negative PET/CT scan had an average follow-up period of 2.8 years, and the median was 2.9 years (range: 0.11–5.39 years).

The sensitivity, specificity, positive predictive value (PPV), and negative predictive value (NPV) were computed with their corresponding Agresti–Coull 95% confidence intervals using SPSS Statistics 28.0 (IBM Corp. Released 2021. IBM SPSS Statistics for Windows, Version 28.0. Armonk, NY, USA: IBM Corp).

### 2.2. Investigation Protocol

All the investigations were performed according to the guidelines of the German Association of Nuclear Medicine. A median F-18-FDG activity value of 218 MBq (range: 187–265 MBq) was applied. In 23 PET/CT examinations (74%), an iodine-containing contrast medium was additionally used, while 8 examinations (26%) were performed without it due to various contraindications (e.g., known allergy).

### 2.3. Analysis

Two independent, board-certified nuclear medicine physicians evaluated all of the PET/CT scans. If there was no consensus, a third experienced nuclear physician was involved in the evaluation. Finally, any result that was approved by two nuclear physicians was included. The physicians had no information about the evaluation of the other diagnosticians, the primary tumor position, or the outcome of the patient.

The study was approved by the Ethics Committee of Marburg University Hospital (Az.: RS 22/12).

## 3. Results

### 3.1. Primary Tumor Lesion

In total, 18 patients (18/31; 58%) were found to have a localized primary tumor during the course of their medical history (Figure 1 and Figure 2).

The most frequent primary tumor location was the oropharynx. A total of twelve primary tumors (12/18; 67%) were located in the oropharyngeal region. Of these, seven (7/18; 39%) were assigned to the base of the tongue and one (1/18; 6%) to the tonsils. In the remaining four patients (4/18; 22%) with a primary tumor in the oropharynx, the exact location was not documented.

Three primary tumors (3/18; 17%) were found outside the oropharynx but in the ENT region. Of these, two tumors (2/18; 11%) were detected in the parotid region and one (1/18; 6%) in the hypopharyngeal region. The TN and HPV status of the histologically confirmed tumors of the ENT area at initial diagnosis are given in Table 2.

In addition, three primary tumors (3/18; 17%) were detected outside the head and neck region: two in the lungs and one in the esophagus.

Of the 18 primary tumors found, 12 (12/18; 67%) were detected by a PET/CT scan, 5 (5/18; 28%) by panendoscopy following a PET/CT scan, and 1 primary tumor (1/18; 6%) by tonsillectomy.

In two patients (2/31; 6%), the area of FDG uptake in the PET/CT scan was not biopsied after the examination; no definitive statement about the final primary tumor location can be established for these patients.

In 11 patients (11/31; 35%), no primary tumor could be found in the course of the patient history; thus, these patients were considered to suffer from a “true” CUP syndrome; 10/11 had a true negative PET/CT, while 1 patient had a false positive scan.

In 48% of patients (15/31), the PET/CT scan showed a suspicious lesion; a subsequent biopsy was performed in ten patients. In all biopsy samples, malignant cells indicative of a primary tumor were found.

Of the five patients who did not undergo biopsy, in two cases, the primary tumor in the area identified by PET/CT was confirmed clinically in the further course of the disease. In two other patients, the location remained unclear because the area of FDG accumulation was not biopsied; these were not considered in the evaluation (the determination of the detection quality of the PET/CT examination related to the primary tumor position). The fifth patient was the only one in this study who showed a false positive result in the PET/CT scan. Retrospectively, the suspected tumor lesion was related to a previously performed panendoscopy with biopsy and was then misinterpreted as the primary tumor.

In 52% of the patients (16/31), there was no suspicious tracer accumulation. This diagnosis was confirmed in ten cases by the end of the follow-up. In six patients, a primary tumor was found despite a negative PET/CT scan.

This results in a sensitivity of 67% (95% confidence interval 43.6–83.9%), specificity of 91%, (95% confidence interval 60.1–100%), a positive predictive value (PPV) of 92% (95% confidence interval 64.6–100%), and a negative predictive value (NPV) of 63% (95% confidence interval 38.5–81.6%), as well as an accuracy of 79% for the F-18-FDG-PET/CT (Table 3).

### 3.2. Prior Diagnostics

In 15 patients (48%) and in 16 foci (52%), suspicion regarding the primary tumor position was raised during the course of diagnostics prior to PET/CT examination. In three patients, the suspicion could be refuted in the course of diagnostics before PET/CT examination by the resection of the lesion. In seven patients, the initial suspicion could be refuted with the help of the PET/CT examination. In three patients, the primary tumor was confirmed by the PET/CT examination at the assumed position by other diagnostic procedures and was confirmed in the course of further medical history.

In two patients, a probable primary tumor position was mentioned in a previously performed diagnostic test that was not detected in the PET/CT scan. The suspicion of pre-diagnosis was confirmed in these two patients during the course of the medical history; thus, these PET/CT scans were evaluated as false negatives.

In one patient, the primary tumor was not found, and the FDG-positive lesion was not biopsied, meaning that a conclusive result regarding the probable primary tumor position prior to PET/CT cannot be obtained.

### 3.3. Distant Metastasis

Altogether, PET/CT examination raised suspicion of distant metastasis in 32% of patients (10/31). Seven (7/31; 23%) of these ten patients had findings that were not previously known; in five (5/31; 16%) of these, seven distant metastases were confirmed histologically or clinically.

### 3.4. Secondary Carcinoma

In 26% (8/31) of the patients, a secondary carcinoma was diagnosed in addition to the CUP in the course of the patient’s history. In one patient, the diagnosis of a secondary carcinoma was firstly indicated by PET/CT examination.

### 3.5. Summary

Regarding all the information obtained by PET/CT, it provided additional information in 24/31 (77%) patients: in 12 patients (12/31, 39%), the primary tumor was found; in 5 patients (16%), a previously unknown metastasis was detected; 1 patient (3%) was diagnosed with a secondary carcinoma; 10 patients (32%) had a true negative result (diagnosis of CUP syndrome); and 7 patients (23%) showed a suspected primary tumor position from a previously refuted diagnosis (Table 4).

## 4. Discussion

The aim of the present study was to analyze the diagnostic value and the real-life performance of FDG-PET/CT in the setting of CUP with cervical lymph node metastasis. Our results show the high impact that PET/CT can have on patients with cervical CUP, due to the possible detection of the primary tumor, metastasis, and/or second malignancies.

There is remarkable heterogeneity (Table 5) in the data of previously published studies [4,5,6,7,8,9]. Comparing our results with these other studies is challenging due to the differences in the extent of diagnostics used prior to PET/CT. A possible explanation for the differing results could be the relatively small patient collectives investigated (range: *n* = 18–*n* = 78), the variable definitions of CUP disease, the different follow-up periods, and a non-uniform understanding of a true-negative PET/CT scan. Nevertheless, the results of the present study align well with those of other publications. Furthermore, the examined collective of patients is similar to the groups of patients examined in the literature, as 84% of the presented patient population were male, 87% between 50 and 80 years of age, and also squamous cell carcinoma was the most common diagnosis (70%) after histopathological lymph node examination. Similarly, cervical lymph node level II was the most commonly reported affected cervical lymph level both in this study and in the literature [10].

A meta-analysis reported the detection of 74 primary tumors in 302 patients with CUP and CLNM using FDG-PET, resulting in a sensitivity of 88.3%, specificity of 74.9%, and accuracy of 78.8% [11]. Nikolova et al. reported a sensitivity of 70% and specificity of 84% for primary tumor detection in patients with CUP and CLNM using PET/CT [12].

Roh et al. were able to reach a sensitivity of 87.5% and specificity of 82.1% in patients without precedent therapy and showed distant metastasis in six of six patients [6]. To achieve comparability, we performed a second analysis, excluding the patients with precedent therapy and a long time lag (>1 year) between the diagnosis of CLNM and PET/CT. For this modified group of patients, FDG-PET/CT reached a sensitivity of 71% and specificity of 86% (Table 3 and Table 5). This led to an increase in sensitivity and a reduction in specificity, and therefore to comparably high values for sensitivity and specificity in our real-life data.

We also performed further analysis comparing the FDG-PET/CT results of patients who did or did not undergo previous diagnostics: the sensitivity, specificity, and positive predictive value were higher for the group without previous diagnostic tests (Table 3). Yabuki et al. reported a sensitivity of 80.8% and specificity of 76.9% for a PET scan in 24 patients with cervical CUP and inconspicuous previous diagnostic procedures (clinical investigations, CT, MRI, panendoscopy), concluding satisfactory statistical values even after previous diagnostics [13].

PET/CT compares favorably to conventional imaging modalities such as CT and MRI in the setting of CUP with CLNM regarding both the identification of primary tumors and the detection of lymph node metastases.

According to the meta-analysis of Burglin et al., PET/CT was able to identify the primary tumor in approximately 41% of cases [14]. This seems superior to reports for CT and MRI, where the rates of the successful localization of squamous cell carcinomas using CT or MRI were 22% and 36%, respectively [15].

Furthermore, the sensitivity of PET/CT (95%) exceeds the sensitivity of MRI (79%) in the detection of lymph node metastases according to Antoch et al. [16], and also appeared to be preferable to CT and MRI in a meta-analysis by Kyzas et al. [17]. Gödény et al. compared FDG-PET/CT to multiparametric MRI in the detection of the primary tumor in patients with CLNM in CUP and recommended PET/CT as the method of choice due to its higher sensitivity and whole-body approach, with the possibility of identifying metastases or second malignancies [18]. Our study highlights these advantages since the suspicion of the localization of a primary tumor expressed in the preliminary examination was refuted by PET/CT in seven patients.

A major benefit of PET/CT, in addition to accurate detection, is the possibility of further evaluation of distant metastasis [18,19]. Distant metastases are associated with an overall poor prognosis; they typically occur in advanced-stage carcinomas. In a study by Al Kadah et al., 10% of the patients with a CUP of the head and neck had distant metastasis [20]. In our study, 10 patients (32%) were suspected to show distant metastases in the PET/CT; in 7 patients (23%), the distant metastases were unknown beforehand. Consequently, PET/CT provided important new information, even in a larger percentage than was to be expected from the literature. Due to the possible identification of metastases, PET/CT plays an important role in tumor staging, but additional multiplanar MRI may be required to examine local tumor invasion [18].

The presence of a second primary tumor in patients with cervical CUP is a crucial prognostic factor since the five-year survival rate in the presence of a second malignancy is only 20–30% [21]. Known noxious agents that promote the development of squamous cell carcinoma include alcohol consumption and nicotine abuse—in particular, the combination of frequent alcohol and concurrent tobacco consumption [22]. Another factor associated with its development is the human papillomavirus (HPV) [23]. Because of this pathogenesis (alcohol, smoking, HPV), the risk of a second primary carcinoma is increased for the entire upper aerodigestive tract. Synchronous second malignancies are found in 5–10% of patients with head and neck squamous cell carcinomas [24,25]. In the presence of a second carcinoma, the management approach is modified and adapted in up to 80% of cases; thus, PET/CT can provide decisive information for further therapy planning [25,26,27].

In conclusion, PET/CT offers a comprehensive diagnostic assessment with regard to the detection of a primary tumor, its staging, and possible second primary tumors simultaneously [28,29].

In order to define a clinically authentic, realistic patient population in regard to achieving our aim of a real-life investigation, the inclusion criteria of this study were intentionally defined “more broadly”. Cervical CUP syndrome is a relatively rare condition, resulting in a relatively small but still representative patient group investigated here. Since this study was carried out as a retrospective analysis, any disadvantages associated with this study design must be mentioned in a critical evaluation of the respective findings. Nevertheless, this study provides important information on a relevant number of cases under authentic circumstances.

## 5. Conclusions

Our real-life analysis highlights the potential of FDG-PET/CT as an accurate imaging modality that enables proper treatment planning for an optimized patient outcome. Furthermore, it is proven to be suitable for the detection of lymph node metastases and distant metastases and also for the detection of second malignancies.

FDG-PET/CT should be considered as a first-line diagnostic tool in CUP with CLNM.

FDG-PET/CT may detect the primary tumor, metastases, and second malignancies.

## Figures and Tables

**Figure 1 biomedicines-10-02095-f001:**
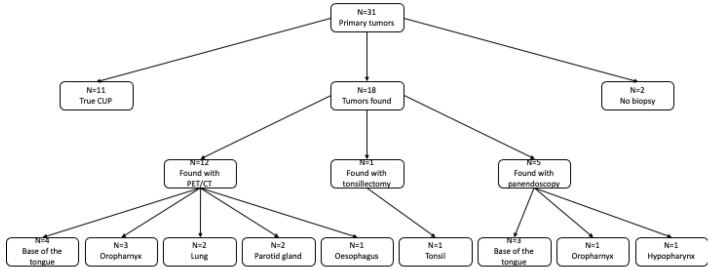
Patient collective.

**Figure 2 biomedicines-10-02095-f002:**
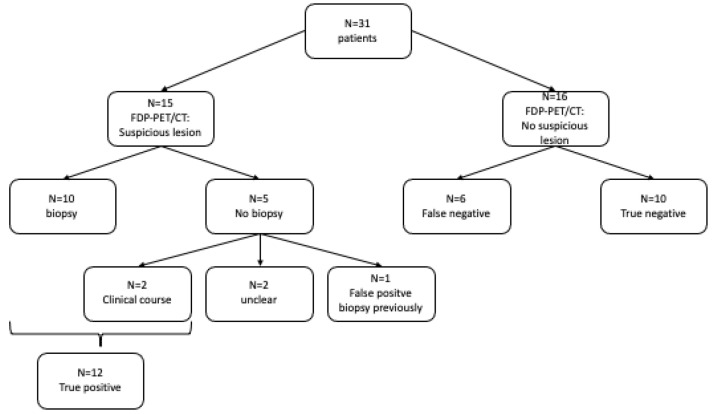
Primary tumor.

**Table 1 biomedicines-10-02095-t001:** Study collective, patient demographics, and diagnostics.

Sex	Female	5 (16%)
	Male	26 (84%)
Age (years)	40–49	3
	50–59	9
	60–69	11
	70–79	7
	80–89	1
Use of alcohol	Yes	12
	No	1
	No information	18
Use of nicotine	Yes	11
	No	3
	No information	17
Lymph node metastasis	Squamous cell carcinoma	20 (65%)
	Dedifferentiated/Not further characterized carcinoma	8 (26%)
	Melanoma	1 (3%)
	Poorly differentiated carcinoma	1 (3%)
	Adenocell carcinoma	1 (3%)
Previous diagnostics	Main diagnostic	15 (48%)
	Minor diagnostic	4 (13%)
	No previous diagnostic	9 (29%)
	No information	3 (10%)
Previous therapy	Radiotherapy	1
	Radiotherapy and chemotherapy	4
	No therapy	26
Timing of PET/CT in relation to diagnosis	Within 1 month	20
	Within 1 year	3
	>1 year	2
	No information	6
	Median time of PET/CT after diagnosis	8 days

Main diagnostic: MRI of the neck, CT of the thorax and neck. Minor diagnostic: Panendoscopy, abdominal diagnostics (abdominal sonography, abdominal CT, diagnostic gastroscopy/colonoscopy), X-ray of the thorax, MRI cranium, dermatological consultation, bronchoscopy, CT cranium.

**Table 2 biomedicines-10-02095-t002:** T and N and HPV status of histologically proven ENT tumors.

T and N and HPV Status of Histologically Proven ENT Tumors *		
T	X	11
	1	9
	2	4
	3	0
	4	2
N	1	6
	2	18
	3	2
HPV	Positive	2
	Negative	3
	Unknown	21

* The three patients with bronchial carcinoma/esophageal carcinoma were excluded, as were the two non-biopsied patients.

**Table 3 biomedicines-10-02095-t003:** PET/CT with regard to previous diagnostic and therapy.

	No Previous Diagnostic	Previous Diagnostic	Patients Excluded with Previous Therapy and Time from Diagnosis to PET > 1 Year	Total Collective
Sensitivity	75% (40.1–93.7%)	63% (30.4–86.5%)	71% (46.6–87.0%)	67% (43.6–83.9%)
Specificity	100% (16.7–100%)	89% (54.3–100%)	86% (46.7–99.5%)	91% (60.1–100%)
Positive predictive value	100% (55.7–100%)	83% (41.8–98.9%)	92% (64.6–100%)	92% (64.6–100%)
Negative predictive value	33% (5.6–79.8%)	73% (42.9–90.8%)	55% (28.0–78.7%)	63% (38.5–81.6%)
Accuracy	87.5%	76%	78.5%	79%

Sensitivity, specificity, positive predictive value, negative predictive value, and accuracy are listed in percent with 95% confidence interval.

**Table 4 biomedicines-10-02095-t004:** Information obtained by F-18-FDG-PET/CT.

F-18-FDG-PET/CT	12/31	Primary tumor lesion
	10/31	True negative (=True CUP syndrome)
	7/31	Diagnosis of the previously refuted diagnostics
	5/31	Unknown metastasis
	1/31	Second primary tumor

Conclusion: New information in 24/31 (77%) patients.

**Table 5 biomedicines-10-02095-t005:** Overview of various studies.

	Patients (*n*)	Sensitivity	Specificity	Positive Predictive Value	Negative Predictive Value
Gutzeit et al. 2005 [4]	18	35%	-	86%	-
Nassenstein et al. 2007 [5]	39	31%	-	73%	-
Roh et al. 2009 [6]	44	88%	82%	74%	92%
Keller et al. 2011 [7]	38	78%	95%	93%	83%
Wong et al. 2012 [8]	78	100%	67%	65%	100%
Lee et al. 2015 [9]	56	69%	88%	88%	69%
Eilsberger et al. 2022 (total collective)	29	67%	91%	92%	63%
Eilsberger et al. 2022 (modified group)	24	71%	86%	92%	55%

## Data Availability

The data that support the findings of this study are available from the corresponding author, F.E., upon reasonable request.

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
