# Peer review of "Real-Life Performance of F-18-FDG PET/CT in Patients with Cervical Lymph Node Metastasis of Unknown Primary Tumor"

_biomedicines, 2022, doi:10.3390/biomedicines10092095_

Round 1

Reviewer 1 Report

Major issues:

F-18-FDG PET/CT use cases have been previously documented well in the literature and the limitations and advantages of this method have been widely discussed. The authors look to use this in a specific context of cervical CUP and share the results. Although the case reports are useful and demonstrates success in a few patients, the claims and conclusions seem overly optimistic (calls excellent accuracy in conclusion) and do not seem to reflect the numbers shared in the study for sensitivity and NPV, in addition accuracy metric was not calculated. The discussion section needs more structuring and re-writing to address concerns and talk about potential limitations of technology with regards to how it can be used, whether or not it can be used for staging.  In addition, the discussion seems to have sentences that seem like heading (“Recent studies confirm these high values for sensitivity, specificity.”)? In that case please change this to heading and further elaborate in writing. Overall only 31 patients were studied and hence sharing confidence intervals for the numbers reported, i.e sensitivity , specificity , NPV and PPV would be beneficial.   

Minor issues:

The authors could reformat some of the flow diagrams (Figure 1), to remove boxes with background so the text and figure quality is improved. 

Author Response

Point 1: F-18-FDG PET/CT use cases have been previously documented well in the literature and the limitations and advantages of this method have been widely discussed. The authors look to use this in a specific context of cervical CUP and share the results. Although the case reports are useful and demonstrates success in a few patients, the claims and conclusions seem overly optimistic (calls excellent accuracy in conclusion) and do not seem to reflect the numbers shared in the study for sensitivity and NPV, in addition accuracy metric was not calculated.

Response to point 1: Thank you very much for this helpful critique. We have calculated and added the accuracy and rephrased the statement.

Point 2: The discussion section needs more structuring and re-writing to address concerns and talk about potential limitations of technology with regards to how it can be used, whether or not it can be used for staging.  In addition, the discussion seems to have sentences that seem like heading (“Recent studies confirm these high values for sensitivity, specificity.”)? In that case please change this to heading and further elaborate in writing.

Response to point 2: We have rewritten the discussion as well as optimized it.

Point 3: Overall only 31 patients were studied and hence sharing confidence intervals for the numbers reported, i.e sensitivity , specificity , NPV and PPV would be beneficial.  

Response to point 3: We have added the required calculations.

Point 4: The authors could reformat some of the flow diagrams (Figure 1), to remove boxes with background so the text and figure quality is improved.

Response to point 4: We have implemented this point of criticism and optimized the flow chart, as well as created another one for a better overview.

Reviewer 2 Report

Authors should be congratulated for their manuscript. However, some issues should be revised before publication.

- English language should be revised by a native speaker

- Organization and appearance of tables should be improved (especially Table 2)

- A figure summarizing the "take-home message" should be provided

Author Response

Point 1: English language should be revised by a native speaker.

Response to point 1: We have used the english editing service of mdpi.

Point 2: Organization and appearance of tables should be improved (especially Table 2).

Response to point 2: We improved the tables and implemented a figure instead of Table 2.

Point 3: A figure summarizing the "take-home message" should be provided

Response to point 3: We provided two take-home messages in a listing

Round 2

Reviewer 1 Report

Some decimal points are represented with comma instead of full stop. Please change this throughout manuscript for improving readability. I would suggest improving clarity in flow diagrams for final print, the figures 1 and 2 seem blurry. 

Author Response

Thank you very much for the optimization suggestions. We have gladly implemented them:

- In some decimal points changed comma to of full stop
- Improving the figures 1 and 2